# LEARNING UNCERTAINTY FOR UNKNOWN DOMAINS WITH ZERO-TARGET-ASSUMPTION

**Yu Yu**[*]    **Hassan Sajjad**[†]    **Jia Xu**[*]

[*]School of Engineering and Science, Stevens Institute of Technology
[†]Faculty of Computer Science, Dalhousie University
`yyu50@stevens.edu, hsajjad@dal.ca, jxu70@stevens.edu`

## ABSTRACT

We introduce our Maximum-Entropy Rewarded Reinforcement Learning (MERRL) framework that selects training data for more accurate Natural Language Processing (NLP). Because conventional data selection methods select training samples based on the test domain knowledge and not on real life data, they frequently fail in unknown domains like patent and Twitter. Our approach selects training samples that maximize information uncertainty measured by entropy, including observation entropy like empirical Shannon entropy, Min-entropy, Rényi entropy, and prediction entropy using mutual information, to cover more possible queries that may appear in unknown worlds. Our MERRL using regularized A2C and SAC achieves up to -99.7 perplexity decrease (-43.4% relatively) in language modeling, +25.0 accuracy increase (+40.0% relatively) in sentiment analysis, and +5.0 F1 score increase (+30.8% relatively) in named entity recognition over various domains, demonstrating strong generalization power on unknown test sets.

## 1 INTRODUCTION

We introduce novel training set selection method that does not require target-domain information to improve out-of-domain Natural Language Processing (NLP) model accuracy. Machine learning is a data-driven process whose success relies highly on the data in use. System performance is typically measured on a specific test set, however, in reality, the test domain is often oblivious during model training, resulting in a critical performance gap between laboratory findings and language use in the real world. For example, we often observe that a system that relies on human parity results generates surprising errors in real-life use scenarios.

Some work has been done in augmenting or selecting data (Wang et al., 2022) to address this discrepancy. Data optimization can be expensive and error-prone for general domains (Jha et al., 2020). Thus, conventional approaches choose critical in-domain data that may work well for a pre-defined target domain (Moore & Lewis, 2010; Kirchhoff & Bilmes, 2014; van der Wees et al., 2017; Fan et al., 2017; Qu et al., 2019; Liu et al., 2019; Kang et al., 2020). However, there are two problems with domain-specific data selection: First, shifting data toward one target domain may fail in the source and other domains. Second, when target domains are unknown, as in the case of most real-world applications, we do not know what future data to receive before model launches.

In our study, we select training data without using target-domain information to achieve learning generalization. Our data selection objective is to maximize the uncertainty of the training data. Specifically, we use entropy to measure the uncertainty based on the principle of maximum entropy, which states that subject to known constraints, the probability distribution that best represents the current state of knowledge is the one with the largest entropy (Jaynes, 1957; Katz, 1967; Hernando et al., 2012). Therefore, a system with the largest remaining uncertainty contains the least extra biases or uncalled-for assumptions and is ideal for modeling distributions for unknown test domains.

To that end, we propose to measure the amount of uncertainty in our observational data and in our model prediction output. As observation entropy, we use Shannon Entropy, Rényi Entropy, and Min Entropy on the $n$-gram relative frequency of all sentences in the dataset instead of one sentence to model the dependency among sentences. As prediction entropy, we compute the mutual information

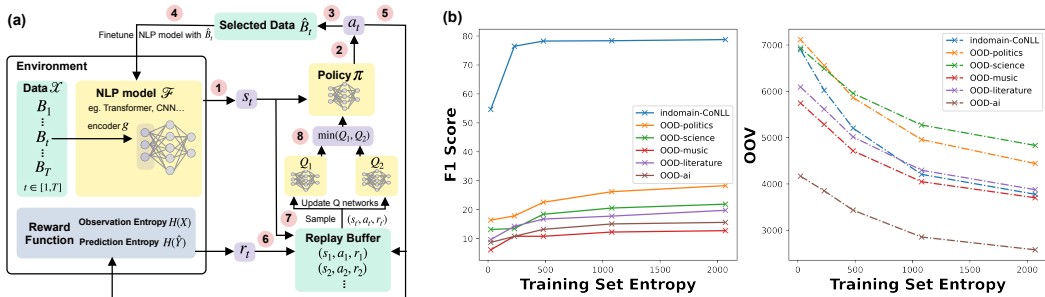

Figure 1: (a): Maximum-Entropy Rewarded Reinforcement Learning framework. (b): Higher training set entropy, better learning generalization, w.r.t. F1 score and OOV.

between the neural network input and its latent representation to quantify how well the information is compressed according to the Information Bottleneck principle. In this way, our approach makes it possible to model inter-dependencies among samples that are critical to improve learning but often neglected (Steinwart et al., 2009; Zhelezniak et al., 2019; Fan et al., 2017).

Putting things into NLP context, we may ask: *"Why does higher entropy of the training dataset lead to a more generalized learning ability of an NLP model?"* Consider a toy example of three sentences {To be. Not to be. To be or not to be.} with frequencies of the words "or" (1), "to" (4), "be" (4), "not" (2). Although "to" occurs more often, "not" represents the opposite meaning and contributes more to the Shannon entropy value. As a hypothetical example, we assume these four words compose the full vocabulary of our world. Now consider that each word is a sample, i.e., $\Pr(\text{"to"}) = \frac{4}{11}$, $\Pr(\text{"or"}) = \frac{1}{11}$, $\Pr(\text{"be"}) = \frac{4}{11}$, and $\Pr(\text{"not"}) = \frac{2}{11}$. Suppose there are subsets $A$ and $B$, where subset $A$ selects "to" four times, which has a unigram entropy of $0.16$, while subset $B$ selects "to", "or", "be", and "not" each one time, which has a unigram entropy of $0.49$. The entropy of subset $B$ is higher than subset $A$, and the (maximum) out-of-vocabulary (OOV) of subset $B$ is smaller than subset $A$ (for a random test), suggesting more generalized data for training that results in more accurate predictions. This observation denotes that increasing the entropy of training data helps build a generalized machine learning model.

Moving from the above hypothetical example, *in a real dataset, does higher entropy also indicate better learning generalization, specifically fewer OOV words, and higher prediction accuracy?* Figure 1-(b) shows our named entity recognition (NER) task results on the CoNLL2003 dataset (Sang & Meulder, 2003) with one in-domain and five out-of-domain (OOD) test sets, details in Appendix. We observe that the unigram entropy of the training subset negatively correlates (Pearson correlation coefficient: $-0.94$) to the OOV of six test sets and strongly positively correlates to the in-domain and out-of-domain test F1 scores (Pearson correlation coefficient: $0.80$). This result indicates that the subset with higher entropy is more likely to generalize on a new test domain with a lower OOV rate and higher F1 score, demonstrating that the training set optimization using entropy can effectively enhance prediction accuracy on unseen domains.

Knowing that a training set with higher entropy leads to more generalized learning, *how can we optimize the subset to maximize the information content without any target domain assumption?* In general, the subset selection optimization problem is computationally intractable, and we use regularized Advantage Actor Critic (A2C) (Mnih et al., 2016) and Soft Actor Critic (SAC) (Haarnoja et al., 2018) to approximate the set optimization. As illustrated in Figure 1-(a), our method equipartitions the training data into mini-batches and simultaneously learns a policy network to select data sequentially and two Q networks to estimate future returns with our entropy rewards. MERRL has the advantages of low variance, monotonic policy improvement, sampling efficiency, and significantly outperforms data selection baselines (Ma et al., 2019; Liu et al., 2019; Aharoni & Goldberg, 2020).

Our work contributes four important components to ongoing work on learning generalization:

1. Maximizing uncertainty measured by entropy for learning generalization without target domain assumptions;

2. Entropy-regularized A2C and SAC reinforcement learning algorithms with entropy rewards for training subset optimization that is typically computational intractable;

3. A data selection framework MERRL by modeling training sample dependency that demonstrates significant improvement in NLP accuracy and generalization on various tasks and domains

The rest of the paper is organized as follows. In Section 2, we introduce the MERRL in detail. Then in Section 3, we empirically verify the generalization and accuracy improvement using MERRL. We discuss related work in Section 4 and conclude the paper in the last section.

## 2   METHOD

Below, we describe our MERRL framework in detail, including problem definitions (Section 2.1), the proposed framework (Section 2.2), the training algorithms (Section 2.3), and the entropy-based reward functions (Section 2.4).

### 2.1   DEFINITIONS

In training set optimization, we formalize the components of the environment as illustrated in Figure 1 (a), including a training dataset, an NLP model $\mathcal{F}$, and a reward function $\mathcal{R}$. The training set is denoted as $\mathcal{X} = \{x_i\}_{i=1}^n$ where $x_i$ is a sentence (document) and $n$ is the training set size. We shuffle and randomly partition $\mathcal{X}$ into $T$ disjoint data batches (Liu et al., 2019) so that $\mathcal{X} = \{\mathcal{B}_t\}_{t=1}^T = \{\mathcal{B}_1, \mathcal{B}_2, ..., \mathcal{B}_T\}$, with $\mathcal{B}_t = \{x_{(t-1)n|T+1}, x_{(t-1)n|T+2}, ..., x_{tn|T}\}$. $n|T$ is the integer division of $n$ by $T$, and $T \leq t$. If $\mod (n, T) \neq 0$, then the last batch has a variable size of $\mod (n, T)$ and collects the remaining sentences.

MERRL selects a subset of data from each mini batch in sequence. The series of selection can be viewed as a sequential decision-making process and can be modeled by a Markov decision process (MDP), consisting of four elements: a set of states $\mathcal{S}$, a set of actions $\mathcal{A}$, a transition function $\mathcal{P} : \mathcal{S} \times \mathcal{A} \times \mathcal{S} \rightarrow [0, \infty)$, and a reward function $\mathcal{R} : \mathcal{S} \rightarrow \mathbb{R}$. Given an MDP $(\mathcal{S}, \mathcal{A}, \mathcal{P}, \mathcal{R})$, the goal of a reinforcement learning system, or an agent, is to learn an optimal policy function $\pi$, which is a mapping from the set of states $\mathcal{S}$ perceived from the environment to a set of actions $\mathcal{A}$, or formally $\pi : \mathcal{S} \rightarrow \mathcal{A}$ (Uc-Cetina et al., 2021). In our data selection context, the MDP elements $(\mathcal{S}, \mathcal{A}, \mathcal{P}, \mathcal{R})$ are specified as: the observation space $\mathcal{S} \in \mathbb{R}^{|\mathcal{B}_t| \times d}$ where $|\mathcal{B}_t|$ is size of a batch and $d$ is the sentence (document) embedding dimension; the action space $\mathcal{A} \in \mathbb{R}^{|\mathcal{B}_t|}$; the uniform transition function $\mathcal{P}$ which gives the next state; and the entropy-based reward functions $\mathcal{R}$ (details in Section 2.4).

### 2.2   MERRL FRAMEWORK

In our reinforcement learning (RL) setting, the policy $\pi$ interacts with the environment over a number of discrete time steps $\mathcal{T}$, and stores the collected experience $(s, a, r)$ into the replay buffer. After some fixed time, the replay buffer samples a tuple and updates the Q networks and policy network respectively. At each time step $t \in \mathcal{T}$, the policy $\pi$ receives a batch of sentence embeddings from the environment and selects a subset of data. Then, the environment gives the next state $s_{t+1}$ and a scalar reward $r_t$ to the agent. The reward $r_t$ measures how good the selected data is. The return is the total discounted accumulated reward $\mathcal{R}_t = \sum_{j=0}^{T-t} \gamma^j r_{t+j}$ from time step $t$ to terminal time step $\mathcal{T}$ with discount factor $\gamma \in [0, 1]$. Our goal is to learn an optimal policy $\pi$ to maximize the expected return from each state $s_t$.

Each time step contains eight steps as shown in Figure 1 (a): At **step 1**, an encoder (e.g. an embedding layer in LSTM, or an encoder in transformer) inside the NLP model transforms the batch of raw data $\mathcal{B}_t$ into a batch of (document) embeddings, denoted as $s_t$. Next, at **step 2** and **step 3**, the policy outputs action $a_t$ along with the selected data $\hat{\mathcal{B}}_t$. Specifically, the policy takes the state $s_t$ as input and outputs a probability distribution for $s_t$, so that each sentence is associated with a probability representing how likely it is going to be selected. The selected subset $\hat{\mathcal{B}}_t$, is then obtained by Bernoulli sampling each sentence in the state $s_t$. The result of Bernoulli sampling is represented as an action vector $a_t$, where each value in it is either $0$ or $1$ representing each sentence in the batch not

being or being selected. At **step 4**, as soon as we obtain $\hat{\mathcal{B}}_t$, the NLP model $\mathcal{F}$ as well as encoder $g$ are finetuned by the selected subset $\hat{\mathcal{B}}_t$. At **step 5**, the scalar reward $r_t = \mathcal{R}(s_t, a_t)$ is calculated by designed reward functions $\mathcal{R}$ (which we give definitions in Section 2.4). Next in **step 6**, the tuple of $(s_t, a_t, r_t)$ is stored in the replay buffer. After some fixed time steps, at **step 7**, we sample a previously stored tuple to update the two Q networks. Finally, at **step 8**, we take the minimum between the outputs of two Q networks given the sampled $(s_{t'}, a_{t'}, r_{t'})$ to update the policy network $\pi$ with regard to the objectives expanded in next section 2.3.

### 2.3 ENTROPY-BASED TRAINING ALGORITHMS

We draw on two algorithms to estimate the policy function $\pi$, the on-policy Advantage Actor Critic (A2C) with entropy regularization and off-policy Soft Actor Critic (SAC).

#### 2.3.1 A2C WITH ENTROPY REGULARIZATION

The A2C algorithm maintains a policy function $\pi_\theta$ and a value function $V_{\theta_v}$. It builds on the vanilla policy gradient method that directly optimizes the policy function by performing gradient ascent on $\nabla_\theta \log \pi(a_t|s_t)\mathcal{R}_t$, which is an unbiased estimate of $\nabla_\theta \mathbb{E}[\mathcal{R}_t]$. Intuitively, it increases the log probability of the sampled action, weighted by the return $\mathcal{R}_t$ (Uc-Cetina et al., 2021). The value function $V_{\theta_v}$ is used as a baseline scaling the policy gradient to reduce the variance in the optimization process with the objective $\mathbb{E}_t\left[(r_t - \mathcal{V}(s_t))^2\right]$ (Schulman et al., 2015; Mnih et al., 2015). With the baseline, the gradient of policy function becomes $\nabla_\theta \log \pi(a_t|s_t)A_t$, where $A_t$ is estimated by the difference between the the empirical return $\mathcal{R}_t$ and value function $\mathcal{V}(s_t)$ as $\sum_{j=0}^{T-t-1} \gamma^j r_{t+j} + \gamma^{T-t}\mathcal{V}(s_T) - \mathcal{V}(s_t)$. To enhance the robustness of the policy in the face of high-dimensional action space, we refer to the maximum entropy objective (Ziebart, 2010) which augments the standard reinforcement learning objective with an entropy term $\mathcal{H}(\pi(\cdot|s_t))$ to encourage exploration of diverse behaviours and stabilize training(Mnih et al., 2016; Schulman et al., 2017). Consequently, the parameters of policy function $\theta$ and value function $\theta_v$ are updated by:

$$\theta_{t+1} = \theta_t + \alpha(\nabla_\theta \log \pi_\theta(a_t|s_t)A_t + \beta\nabla_\theta\mathcal{H}(\pi(s_t;\theta))) \tag{1}$$

$$\theta_{v(t+1)} = \theta_{vt} - \alpha\nabla_{\theta_{vt}}(r_t - \mathcal{V}(s_t))^2 \tag{2}$$

where $\alpha$ is learning rate, $\mathcal{H}$ is the entropy of policy $\pi$, and $\beta$ controls the trade-off between exploitation and exploration.

#### 2.3.2 SAC

Though A2C with maximum entropy objective improves the stability of training, it suffers from poor sample efficiency. In contrast, SAC (Haarnoja et al., 2018) uses a replay buffer to reuse past experiences to reduce sample complexity. To this end, SAC maintains a soft Q-function $Q_\phi(s_t, a_t)$ and a policy function $\pi_\theta(a_t, s_t)$, where $\phi$ and $\theta$ are the parameters for these networks respectively. The soft Q-function parameters can be optimized with the objective of soft Bellman residual:

$$J_Q(\phi) = \mathbb{E}_{(s_t, a_t)\sim D}\left[\frac{1}{2}(Q_\phi(s_t, a_t) - (r(s_t, a_t) + \gamma\mathbb{E}_{s_{t+1}\sim p}\left[V_{\bar{\phi}}(s_{t+1})\right]))^2\right] \tag{3}$$

where the parameters $\bar{\phi}$ are obtained as an exponentially moving average of $\phi$, and the soft state value function $V$ is defined as below following the SAC for discrete action settings (Christodoulou, 2019):

$$V(s_t) = \pi(s_t)^T\left[Q(s_t) - \beta\log(\pi(s_t))\right] \tag{4}$$

The policy parameters are updated towards the exponential of the new Q-function with the KL-divergence objective, and it can be further transformed to the following form for the discrete action settings:

$$J_\pi(\theta) = \mathbb{E}_{s_t\sim D}\left[\pi_t(s_t)^T\left[\beta\log(\pi_\theta(s_t)) - Q_\phi(s_t)\right]\right] \tag{5}$$

In practice, SAC maintains two soft Q-functions $Q_{\phi 1}$ and $Q_{\phi 2}$ and substitutes the soft Q-functions in equation 3 and equation 5 with $\min(Q_{\phi 1}, Q_{\phi 2})$ to mitigate bias (Fujimoto et al., 2018).

## 2.4 Entropy-based reward functions

We introduce two classes of reward functions from the angle of syntactic heuristics of training data (2.4.1) and the theory of information bottleneck (2.4.2).

### 2.4.1 Observation Entropy

Although there is no consensus on what are the best in-domain data for generalization, experiments (Adila & Kang, 2022) find models latch on syntactic heuristics, like the overlap of words between in-domain and out-of-distribution sentences to make predictions. Ruder & Plank (2017) demonstrates extracting word entropy as heuristic features to select training data favors domain adaptation in NLP. Based on these findings, we follow classic count-based methods (Song et al., 2012; Ruder & Plank, 2017; Parcheta et al., 2018; Tevet & Berant, 2020), or $N$-grams, as an indicator of how good the selected data is. Specifically, we apply Shannon Entropy (Shannon, 1948), Rényi Entropy (Rényi, 1961) and Min Entropy (Smith, 2011) as reward functions in our reinforcement learning framework. All entropy measures are computed on word $n$-gram relative frequency on all sentences in the dataset.

For a set $G$ with $M$ sentences, and each sentence $x_i$ containing $J_i$ words, we define the *empirical set entropy* as the sum of $n$-gram entropy:

$$\mathcal{H}(G) \;=\; \sum_{i=1}^{M} h(x_i; n)$$

$$h(x_i; n) \;=\; \frac{1}{\alpha - 1} \log p(x_{ij}^{j+n-1})^\alpha,$$

where $p(x_{ij}^{j+n-1})$ is the relative frequency of $n$-gram from word $j$ to $j+n-1$ of sentence $x_i$, and $\alpha$ is a parameter of **Rényi Entropy** controlling the order of entropy. Especially, when $\alpha$ approaching 1, Rényi Entropy is equivalent as **Shannon Entropy**; when $\alpha$ approaching infinity, Rényi Entropy converges to **Min Entropy**.

For a set $G$ with $M$ training examples, we define the *k-th order interpolated set entropy* as a linear combination of $n$-gram entropy from $n = 1$ until $n = k$, weighted $\lambda$, where $\sum_{n=1}^{k} \lambda_n = 1$. For example, if $k = 3$, then it combines unigram, bigram, and trigram set entropy with weight $\lambda_1$, $\lambda_2$, and $\lambda_3$, respectively:

$$\mathcal{H}'(G) = \sum_{n=1}^{k} \lambda_n \sum_{i=1}^{M} h(x_i; n), \tag{6}$$

We use the *2-nd order interpolated set entropy* as our default setting in the following sections.

### 2.4.2 Prediction entropy

From the information theory perspective, the Information Bottleneck (IB) principle indicates the mutual information between the input of a neural network and its latent representation needs to be well-compressed to generalize well on out-of-domain data (Tishby et al., 2000; Tishby & Zaslavsky, 2015). Specifically, IB seeks to obtain a latent representation $\mathcal{Z}$ such that the mutual information between input $\mathcal{X}$ and $\mathcal{Z}$, denoting as $\mathcal{I}(\mathcal{X}; \mathcal{Z})$ is minimized, and the mutual information between $\mathcal{Z}$ and output $\mathcal{Y}$, denoting as $\mathcal{I}(\mathcal{Y}; \mathcal{Z})$, is maximized. Formally, IB is implemented by minimizing the following Lagrangian:

$$minimize\{\mathcal{I}(\mathcal{X}; \mathcal{Z}) - \lambda \mathcal{I}(\mathcal{Y}; \mathcal{Z})\} \tag{7}$$

Intuitively, the smaller mutual information $\mathcal{I}(\mathcal{X}; \mathcal{Z})$ is, the better $\mathcal{Z}$ compresses $\mathcal{X}$, the less likely $\mathcal{Z}$ learns spurious correlations with $\mathcal{X}$, the more robust representation $\mathcal{Z}$ is. However, since $\mathcal{Z}$ is high dimensional, the exact computation of mutual information $\mathcal{I}(\mathcal{X}; \mathcal{Z})$ is intractable. Following Zhao et al. (2020) that bounds $\mathcal{I}(\mathcal{X}; \mathcal{Z})$ with the prediction entropy $\mathcal{H}(\hat{\mathcal{Y}})$, we approximately minimize $\mathcal{I}(\mathcal{X}; \mathcal{Z})$ by empirically calculating the prediction entropy $\mathcal{H}(\hat{\mathcal{Y}})$:

| | auto | beauty | food | instruments | office | computer | tools | phones | grocery | jewelry | outdoor | avg |
|---|---|---|---|---|---|---|---|---|---|---|---|---|
| ALL | 56.48 | 54.03 | 56.02 | 59.15 | 56.84 | 55.17 | 55.95 | 52.79 | 53.06 | 56.63 | 55.81 | 55.63 |
| RAND | 57.07 | 54.93 | 56.93 | 59.55 | 59.28 | 56.16 | 58.04 | 53.77 | 54.81 | 57.92 | 57.30 | 56.89 |
| MTL | 59.46 | 55.89 | 60.73 | 61.88 | 61.87 | 56.20 | 62.50 | 53.93 | 57.86 | 58.98 | 58.23 | 58.86 |
| PLM | 63.45 | 52.35 | 68.37 | 56.09 | 75.48 | 34.54 | 22.32 | 51.58 | 49.06 | 76.09 | 56.17 | 55.04 |
| COS | 53.94 | 54.09 | 68.66 | 63.90 | 63.22 | 41.34 | 41.96 | 45.33 | 68.11 | 65.70 | 57.12 | 56.67 |
| VPG | 73.43 | 62.97 | 77.31 | 79.42 | 78.41 | 63.04 | 81.25 | 61.26 | 70.78 | 73.29 | 70.11 | 71.94 |
| A2C-OE | 77.79 | 65.60 | 81.87 | 84.30 | 83.02 | 67.86 | 86.61 | 62.16 | 72.93 | 76.75 | 74.43 | 75.82 |
| A2C-PE | 78.39 | 66.66 | 81.20 | 83.96 | 82.07 | 67.88 | 86.92 | 62.69 | 73.80 | 76.71 | 75.03 | 75.95 |
| SAC-OE | 78.39 | 66.83 | 81.58 | 83.43 | 82.21 | 68.37 | **87.50** | 61.01 | 73.36 | 76.85 | 75.03 | 75.87 |
| SAC-PE | **78.53** | **67.00** | **81.98** | **84.37** | **83.96** | **68.46** | **87.50** | **62.99** | **73.95** | **76.95** | **75.22** | **76.19** |
| +% | 19.07 | 11.11 | 21.02 | 22.26 | 22.09 | 11.65 | 25.00 | 9.06 | 14.84 | 17.49 | 16.99 | 17.33 |

Table 1: Sentiment analysis accuracy [%] on amazon unprocessed domains. Baselines **PLM** (Ma et al., 2019), **COS** (Aharoni & Goldberg, 2020) and **VPG** (Liu et al., 2019) use test/target domain data of each column, while our methods outperform all of them without using any target domain knowledge. Last row: absolute improvement between **SAC-PE** and best domain generalization method **MTL** (Blanchard et al., 2021)

$$\mathcal{I}(\mathcal{X};\mathcal{Z}) \geq \mathcal{I}(\mathcal{X};\hat{\mathcal{Y}}) = \mathcal{H}(\hat{\mathcal{Y}}) - \mathcal{H}(\hat{\mathcal{Y}}|\mathcal{X}) = \mathcal{H}(\hat{\mathcal{Y}}) \tag{8}$$

$$\mathcal{H}(\hat{\mathcal{Y}}) \approx -\frac{1}{n}\sum_{i=1}^{n}\sum_{j=1}^{|\mathcal{Y}|} p_j(x_i;\theta)\log p_j(x_i;\theta) \tag{9}$$

where $p_j(x_i;\theta)$ is the predicted probability of label $\mathcal{Y}_j$ of sample $x_i$, given the model $\theta$, and $|\mathcal{Y}|$ is the set of all labels. Adopting this observation into our context, we minimize $\mathcal{I}(\mathcal{X};\mathcal{Z})$ using $-\mathcal{H}(\hat{\mathcal{Y}})$ as the reward to select training data within a mini-batch that can learn the optimal latent representation for out-of-distribution generalization.

## 3 EXPERIMENTS

We describe our experimental details and demonstrate that MERRL improves baselines in three NLP applications among various out-of-distribution domains, including two classification tasks, sentiment analysis, named entity recognition, and one generation task of language modeling, without any out-of-domain knowledge. For each task, we experiment with two reinforcement learning algorithms to train the data selector, as well as three reward functions, i.e. **A2C-OE** denotes A2C with entropy regularization rewarded by Observation Entropy. We give a list of hyperparameters used in MERRL in appendix A.2.

### 3.1 NLP EXPERIMENTS

**Baselines** We compare our methods with six baselines: 1) **ALL** The models are trained on all in-domain training data; 2) **RAND** The models are trained on randomly selected 50% in-domain data; 3) **MTL** Marginal transfer learning by Blanchard et al. (2021), a domain generalization framework using kernel methods to augment feature space. 4) **PLM** (Ma et al., 2019) that uses the large pretrained language model BERT (Devlin et al., 2018b) to learn a domain classifier and select data according to probability given by the domain classifier. 5) **COS** (Aharoni & Goldberg, 2020) that uses cosine distance to measure the distance between an in-domain sentence and the centroid of a target domain (out-of-distribution domain), and select sentences close to the target domain. 6) **VPG** (Liu et al., 2019) that uses the vanilla policy gradient method to choose data from a target distribution that resembles in-domain distribution. To be noted, **PLM**, **COS** and **VPG** are all data selection methods requiring the usage of out-of-domain data, while **ALL**, **RAND**, **MTL** and all our methods do not use any out-of-domain knowledge. For the training data size, **ALL** and **MTL** use all in-domain training data only; **VPG** and our methods choose roughly 50% in-domain training data (complete data statistics in Appendix A.4), and we control **PLM** & **COS** which both require a pre-defined selected data size to select 50% in-domain data.

**Sentiment Analysis** We use the Amazon product review dataset (Blitzer et al., 2007) for the sentiment analysis task. Specifically, we use the processed labeled domain data (books, dvd and kitchen) to train our task model and the unprocessed 21 domains as test data. We use a CNN classifier (Kim,

| | politics | science | music | literature | AI | | WikiText-2 IWSLT'17 | WikiText-2 Bio'21 | Penn Treebank IWSLT'17 | Penn Treebank Bio'21 |
|---|---|---|---|---|---|---|---|---|---|---|
| ALL | 26.49 | 19.84 | 12.26 | 16.38 | 13.92 | | IWSLT'17 | Bio'21 | IWSLT'17 | Bio'21 |
| RAND | 26.07 | 19.68 | 12.81 | 17.47 | 13.68 | ALL | 328.23 | 254.64 | 147.03 | 117.17 |
| MTL | 28.47 | 22.47 | 13.49 | 18.97 | 15.68 | RAND | 515.22 | 456.78 | 234.14 | 157.39 |
| PLM | 26.81 | 22.31 | 14.04 | 18.29 | 16.00 | PLM | 554.56 | 441.29 | 233.80 | 154.30 |
| COS | 28.66 | 20.94 | 12.99 | 19.05 | 14.49 | COS | 410.15 | 314.81 | 190.54 | 169.16 |
| VPG | 28.74 | 22.65 | 12.68 | 19.24 | 15.81 | VPG | 311.76 | 229.53 | 143.45 | 88.86 |
| A2C-OE | 29.29 | 23.48 | 14.32 | 20.17 | 16.02 | A2C-OE | 228.96 | 152.38 | 134.87 | 69.33 |
| A2C-PE | 29.52 | 23.34 | 15.02 | 21.07 | 16.11 | A2C-PE | 186.09 | 131.66 | 132.64 | 73.08 |
| SAC-OE | 29.80 | 23.91 | 15.35 | 21.22 | 16.78 | SAC-OE | 198.47 | 137.85 | 137.03 | 70.03 |
| SAC-PE | **29.90** | **23.95** | **15.42** | **21.43** | **16.95** | SAC-PE | **182.17** | **129.78** | **130.48** | **69.15** |

Table 2: Left: NER F1-scores. Right: Language modeling perplexity scores on two test domains. First row: source training domain; Second row: test domains. Results are averaged over three runs.

| | camera | comptr | magazns | video | toys | train |
|---|---|---|---|---|---|---|
| VPG | 4543 | 5123 | 4399 | 8207 | 3826 | 28908 |
| SAC-OE | 4490 | 5076 | 4298 | 8108 | 3739 | 29538 |

Table 3: OOV of VPG selected data (Liu et al., 2019) and SAC selected data on test domains of amazon product review dataset. Last column: training vocab of the selected set.

2014) as the sentiment analysis model and pre-train the CNN classifier for two epochs following Liu et al. (2019) for a fair comparison. Table 1 shows the results averaged over five random seeds. Our methods outperform all baselines on all unprocessed amazon domains. It is worth noting that even with the test domain knowledge, baselines PLM or COS fail to select the "right" data for specific domains (e.g. tools, computer), on the contrary, our methods can indiscriminately select effective data for all domains. **SAC-PE** gains an average improvement of 17.33 over the domain generalization baseline **MTL** which does not use out-of-domain knowledge either.

**Named Entity Recognition**   We use the CoNLL2003 English NER dataset (Sang & Meulder, 2003) as an in-domain training set and the five domains from CrossNER dataset (Liu et al., 2020) as test sets, which has specialized entity categories for each domain. We finetune the pretrained BERT model (Devlin et al., 2018a) on source training set by adding a linear layer on top of the hidden-states output of the last layer and then report the F1-scores on five test sets in left of Table 2. SAC outperforms A2C across all domains and **SAC-PE** improves the test score on *music* domain up to 14.3% compared to **MTL**.

**Language Modeling**   We experiment with two moderate size datasets WikiText-2 (Merity et al., 2016) and Penn Treebank. Our baseline is a Transformer language model (Vaswani et al., 2017) trained with default hyper-parameters from scratch. The RL loop in Figure 1-(a) initializes the language model to be the checkpoint of the pre-trained transformer model. As for evaluation, we report perplexity scores on datasets from different domains, the English side of IWSLT'17 (TED talk) and the English side of WMT Biomedical'21. The baseline transformer model and all language models trained on selected data are updated by the fairseq toolkit (Ott et al., 2019) and stopped until the in-domain validation perplexity score does not improve for 5 epochs. The evaluation results are shown on the right of Table 2. The perplexity on two test domains has been largely improved and there is at most 43.4% (decrease from VPG-229.53 to SAC-PE-129.78) relative improvement in the biomedical domain with WikiText-2 as in-domain data.

## 3.2   ANALYSIS

**SAC VS. A2C**   We plot the learning curves of three reinforcement learning algorithms on the left of Figure 2. The average reward of SAC is significantly higher than A2C with entropy regularization (shortened as A2C) and VPG. SAC and A2C both converge at around 10000 timesteps, while VPG converges at around 20000 timesteps. Comparing A2C and VPG, it is clearly shown that A2C has a smaller variance than VPG. In short, SAC is the most effective algorithm among the three, and A2C can reduce variance compared to VPG. Particularly, with a limited training time budget (e.g. training with 5000 timesteps), SAC can lead to the best performance in training set optimization, which matches our empirical results.

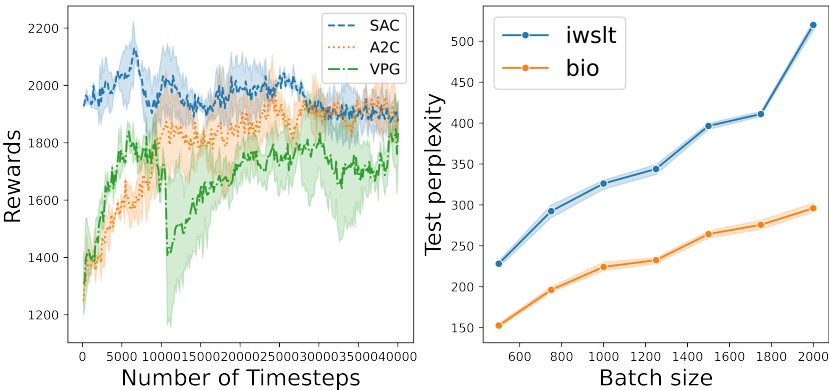

Figure 2: Left: Learning curve of three reinforcement learning algorithms in three random seeds in the NER task. Right: Smaller batch size $|B_t|$ results in better test perplexity on two test sets.

Figure 3: 2-D visualization of sentence embeddings of the selected dataset by VPG (Liu et al., 2019) (blue) and SAC-OE (red). Middle: SAC-OE covers more area of the test domain, e.g. yellow area; Right: SAC-OE has a larger convex hull volume than VPG.

**Time**    In theory, training MERRL and training an NLP model by the selected data takes $\mathcal{T}_{budget}/\mathcal{T}$ times compared to simply training an NLP model using all data from scratch, where $\mathcal{T}$ is the time steps in Markov Decision Process, or the number of batches, and $\mathcal{T}_{budget}$ is a pre-defined hyper-parameter of reinforcement learning training steps. Intuitively, $\mathcal{T}_{budget}/\mathcal{T}$ is the number of epochs in MERRL training, and larger $\mathcal{T}_{budget}$ or smaller $\mathcal{T}$ will lead to longer training time. In practice, training with all in-domain data in sentiment analysis takes 131 seconds while selecting data with SAC-OE takes 1394 seconds ($\mathcal{T}_{budget} = 2000$ and $\mathcal{T} = 140$) on one Tesla V100 GPU, which is roughly ten times faster than the baseline VPG to achieve similar average reward. That being said, the problem of computational cost for MERRL remains to be solved and we leave this as future work.

**Batch size**    Unlike previous applications of reinforcement learning in NLP (Yoon et al., 2020; Fang et al., 2017; Wu et al., 2018) which give reward to a single sample/sentence, our reward function measures the informative level of a whole set of data. In this case, the observation (state) space is no longer a vector, but a batch of vectors. Thus, the batch size $|B_t|$ is a newly introduced hyper-parameter in our subset optimization problem that affects both action space and state space. While previous work uses larger batch size ($|B_t| \geq 2000$) to improve the stability of reinforcement learning training (Yoon et al., 2020; McCandlish et al., 2018), we find that training set optimization can benefit from smaller batch size than large batch size when total training step $\mathcal{T}_{budget}$ is fixed, as shown in the right of Fig 2. The reason can be related to our designed state space, which is not a single vector but a batch of vectors so that a larger batch size can directly enlarge the action space into $2^{|B_t|}$ and make the training harder.

**Visualization**    We plot the t-SNE 2D visualizations using the data selected from the training source domains (books, DVD and kitchen) by VPG (Liu et al., 2019) (blue) and by SAC-OE (red), as well as a surprising (unknown) test domain (magazines, green dots). We embed each sentence using

the sentence-transformer tool (Reimers & Gurevych, 2019). In Figure 3, the middle plot shows the coverage of selected data from SAC-OE (3361 sentences, including 53.3% sentences not overlapping with VPG). While similar in dataset size, blue dots are more densely spread, especially the several dense clusters formed by blue points in the bottom part of the left plot. On the contrary, red dots cover more test domain areas than blue dots, especially in those yellow highlighted areas. To gain more intuition, we draw the convex hull (Barber et al., 2013) for red dots and blue dots respectively, shown on the right side. The red circle encloses the blue circle after removing the outliers from both sets. Furthermore, we compute the out-of-vocabulary of all test domains in the Amazon product review dataset and in-domain vocabulary size of VPG and SAC-OE selected set. In Table 3, SAC-OE has a significantly lower OOV size among all test domains, while more in-domain vocabulary than VPG. In summary, we infer SAC-OE selected data has a superior generalization ability than VPG since it is capable of selecting a training set with a more diverse vocabulary and wider coverage of semantic space.

## 4 RELATED WORK

There have been a number of influential work Moore & Lewis (2010); Axelrod et al. (2011); Ruder & Plank (2017) on data selection that significantly contributed to today's NLP state-of-the-arts. More recently, Fan et al. (2017), Feng et al. (2018), Qu et al. (2019), Fang et al. (2017) and Liu et al. (2019) incorporate reinforcement learning with data selection. Another direction examines the potential of large pretrained language models to select data Yuan et al. (2020); Aharoni & Goldberg (2020); Ma et al. (2019). These work mainly select the training data close to a given target domain for domain adaptation. In contrast, we aim to enhance the model generalization and increase the accuracy on any arbitrary domain. Furthermore, we advance the existing data selection techniques using A2C and SAC that simultaneously optimize the value (Q) network and the policy network for better convergence and lower variance, resulting in higher prediction accuracy and generality.

Beyond the field of NLP, data selection Killamsetty et al. (2021); Durga et al. (2021), data augmentation Volpi et al. (2018); Zhang et al. (2020); Zhou et al. (2020) and data generation Ruiz et al. (2018); Qiao et al. (2020); Mishra et al. (2022) have been widely used in vision, medical and general regression tasks. These methods either utilize augmentation operations on images (e.g, flipping, scaling and cropping), or focus on one specific goal (i.e. image style transfer), or include generative models for adversarial training. Thus, it requires further consideration on how to generalize text data when adapting these methods to NLP tasks. Our method puts an emphasis on both characteristics of text data and general prediction entropy which could be directly generalized to other fields.

Another relevant and emergent line of work is *data pruning*, which aims at selecting a minimum subset of training data to reduce training costs Sorscher et al. (2022); Yang et al. (2022), or to enhance the robustness of the model Kaufmann et al. (2022).

## 5 CONCLUSION

We introduce Maximum-Entropy Rewarded Reinforcement Learning (MERRL) with observation entropy and prediction entropy to select effective training data that significantly enhances the generalization capability of NLP models. We performed experiments using sentiment analysis, named entity recognition and language modeling across various domains. Without any knowledge of out-of-distribution domains, our method outperforms the CNN, BERT and transformer baselines. Our experimental results show that modeling sample dependency by increasing data uncertainty enhances learning generalization and prediction accuracy.

## ACKNOWLEDGMENTS

We appreciate Amazon Alexa Prize, National Science Foundation (NSF) Award No. 1747728, and NSF CRAFT Award No. 22001 to fund this research.

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

## A APPENDIX

### A.1 NOTATIONS

### A.2 HYPERPARAMETERS OF MERRL

### A.3 DETAILS OF THE NER TASK IN INTRODUCTION

We sort 14K training sentences in descending order of entropy and equipartition them into five subsets, where the first subset has the highest unigram set entropy (2069.4), the second subset has the second highest entropy (1078.8), and so on. We then finetune BERT model (Devlin et al., 2018a) on each subset, and compute the F1 scores on six test sets: in-domain CoNLL test set (indomain-test), five out-of-distribution domains in CrossNER dataset (Liu et al., 2020): politics (test1), science (test2), music (test3), literature (test4) and AI (test5).

| Notation | Meaning |
|---|---|
| $\mathcal{F}$ | NLP model |
| $\mathcal{X}$ | Training data set |
| $x_i$ | sentence |
| $|\mathcal{X}|$ | Training set size |
| $T$ | disjoint batch index |
| $T$ | maximum training steps in an episode (epoch) |
| $B_t$ | batch |
| $\hat{B}_t$ | selected batch |
| $g$ | encoder |
| $s_t$ | batch state |
| $s_k$ | single sentence state |
| $a_t$ | action on batch $B_t$ at time step $t$ |
| $a_k$ | action on single sample $x_k$ at time step $t$ |
| $|B_t|$ | batch size |
| $\pi$ | policy |
| $r_t$ | reward at time step $t$ |
| $R_t$ | total future reward from $t$ to $T$ |
| $Q^\pi(s_t, a_t)$ | action value |
| $V^\pi(s_t)$ | expected total future return following $\pi$ since time step $t$ |
| $b(s_t)$ | baseline function |
| $\theta$ | parameters of policy network |
| $\nabla_\theta$ | $\frac{\partial X}{\partial \theta}$ |
| $\nabla_\theta \mathcal{J}(\theta)$ | objective function of policy |
| $L$ | epoch number |
| $\alpha$ | learning rate |
| $\gamma$ | discount factor |
| $\mathcal{F}_I$ | pretrained task model (including encoder $g$) |
| $\mathcal{E}$ | episode record, including $s_t, a_t, r_t$ |
| $G$ | a set of samples |
| $d$ | cosine distance of embeddings |
| $M$ | number of sentences in a set $G$ |
| $J$ | sentence length |
| $h(\cdot; n)$ | $n$-gram entropy |

Table 4: Notation table

| Hyperparameter | Value |
|---|---|
| learning rate | $7e-4$ |
| discount factor | 0.99 |
| entropy coefficient | 0.001 |
| value function coefficient | 0.5 |
| RMSProp epsilon | $1e-5$ |
| number of steps ($T_{budget}$) | 10000 |
| batch size (NER) | 100 |
| batch size (sentiment) | 500 |
| batch size (language modeling) | 500 |

Table 5: Hyperparameters of MERRL

## A.4 $N$-GRAM ENTROPY ALGORITHM

See algorithm 1.

---

**Algorithm 1** $N$-gram set entropy

---

**Input:** Dictionary $d_{uni}, d_{bi}, d_{tri}$ that stores unigram entropy, bigram entropy, trigram entropy for all samples in source training set, a batch of training samples $G = \{(s_i)_{i=1}^M\}$ with size $M$; ratio $\alpha, \beta, \gamma \in [0, 1)$.
**Output:** Reward value of set $N$-gram entropy $H(G)$
1: Initialize $H(G) = 0$;
2: Initialize unigram set entropy, bigram set entropy, trigram set entropy: $h_1(G) = 0, h_2(G) = 0, h_3(G) = 0$;
3: **for all** $s \in G$ **do**
4:     Obtain sentence entropy $d_{uni}[s], d_{bi}[s], d_{tri}[s]$, for $s$;
5:     Update unigram set entropy, bigram set entropy, trigram set entropy:
        $h_1(G) = h_1(G) + d_{uni}[s]$
        $h_2(G) = h_2(G) + d_{bi}[s]$
        $h_3(G) = h_3(G) + d_{tri}[s]$;
6: **end for**
7: $H(G) = \alpha h_1(G) + \beta h_2(G) + \gamma h_3(G)$;
8: return $H(G)$

---

## A.5 DATA STATISTICS

See table 6.

| Task | Source | Method | Selected |
|------|--------|--------|----------|
| Sentiment | 6000 (Amazon) | A2C-RE | 3023 |
| | | A2C-SE | 3287 |
| | | A2C-ME | 3052 |
| | | SAC-RE | 3019 |
| | | SAC-SE | 3361 |
| | | SAC-ME | 3102 |
| NER | 14040 (CoNLL2003) | A2C-RE | 7436 |
| | | A2C-SE | 7764 |
| | | A2C-ME | 7208 |
| | | SAC-RE | 6974 |
| | | SAC-SE | 7125 |
| | | SAC-ME | 7225 |
| LM | 36718 (wikiText-2) | A2C-RE | 18207 |
| | | A2C-SE | 18068 |
| | | A2C-ME | 18135 |
| | | SAC-RE | 18230 |
| | | SAC-SE | 18329 |
| | | SAC-ME | 18329 |
| | 42068 (pennTreebank) | A2C-RE | 21156 |
| | | A2C-SE | 21120 |
| | | A2C-ME | 20956 |
| | | SAC-RE | 20969 |
| | | SAC-SE | 21043 |
| | | SAC-ME | 21157 |

Table 6: MERRL selected data statistics

## A.6 OOV OF MERRL SELECTED DATA

We show the full result of OOV of selected data in table 7.

| Domain | OOV of VPG | OOV of SAC-RE |
|--------|-----------|---------------|
| apparel | 2836 | 2784 |
| auto | 1611 | 1579 |
| baby | 3273 | 3246 |
| beauty | 2965 | 2946 |
| camera | 4543 | 4490 |
| phones | 2178 | 2164 |
| computer | 5123 | 5076 |
| food | 2460 | 2437 |
| grocery | 2180 | 2104 |
| health | 4041 | 4007 |
| jewelry | 1635 | 1618 |
| magazines | 4399 | 4298 |
| music | 9033 | 8975 |
| instruments | 844 | 824 |
| office | 949 | 939 |
| outdoor | 2329 | 2293 |
| software | 4772 | 4740 |
| sports | 4563 | 4523 |
| tools | 151 | 141 |
| toys | 3826 | 3739 |
| video | 8207 | 8108 |

Table 7: Out-of-vocabulary of VPG-selected data Liu et al. (2019) and SAC-selected data on test domains of amazon product review dataset.

| Domain | All | Rand | Threshold | Mtl | PLM | COS | VPG | A2C-SE | A2C-RE | A2C-ME | SAC-SE | SAC-RE | SAC-ME |
|---|---|---|---|---|---|---|---|---|---|---|---|---|---|
| apparel | 49.43 | 50.47 | 50.77 | 47.65 | 51.05 | 51.25 | 51.07 | 50.62 | 50.95 | 50.30 | **51.66** | 50.27 | 50.85 |
| auto | 56.48 | 57.07 | 60.35 | 59.46 | 63.45 | 53.94 | 73.43 | 77.79 | 78.39 | 76.58 | 78.39 | **78.53** | 77.49 |
| baby | 50.42 | 51.09 | 48.18 | 50.80 | 52.48 | 50.90 | 52.81 | 52.51 | 52.55 | 52.37 | 52.66 | **53.16** | 52.95 |
| beauty | 54.03 | 54.93 | 57.33 | 55.89 | 52.35 | 54.09 | 62.97 | 65.60 | 66.66 | 65.38 | 66.83 | **67.00** | 65.61 |
| camera | 49.82 | 50.30 | 50.47 | 50.05 | 50.04 | 50.65 | 49.98 | 50.02 | 50.08 | 49.65 | 49.86 | **51.10** | 50.65 |
| phones | 52.79 | 53.77 | 56.79 | 53.93 | 51.58 | 45.33 | 61.26 | 62.16 | 62.69 | 61.27 | 61.01 | **62.99** | 61.57 |
| computer | 55.17 | 56.16 | 57.80 | 56.20 | 34.54 | 41.34 | 63.04 | 67.86 | 67.88 | 66.25 | 68.37 | **68.46** | 66.73 |
| food | 56.02 | 56.93 | 65.64 | 60.73 | 68.37 | 68.66 | 77.31 | 81.87 | 81.20 | 79.74 | 81.58 | **81.98** | 80.50 |
| grocery | 53.06 | 54.81 | 62.13 | 57.86 | 49.06 | 68.11 | 70.78 | 72.93 | 73.80 | 71.88 | 73.36 | **73.95** | 76.27 |
| health | 50.92 | 49.87 | 50.05 | 49.95 | 50.27 | 50.05 | 49.47 | 49.68 | 49.68 | 49.65 | 49.70 | **51.10** | 49.85 |
| jewelry | 56.63 | 57.92 | 68.18 | 58.98 | 76.09 | 65.70 | 73.29 | 76.75 | 76.71 | 75.57 | 76.85 | **76.95** | 76.27 |
| magazines | 50.78 | 50.56 | 50.76 | 50.42 | 50.53 | 50.30 | 50.44 | 50.55 | 50.97 | 50.72 | 50.94 | **51.94** | 51.44 |
| music | 50.05 | 50.27 | 50.85 | 50.08 | 49.50 | 50.07 | 50.22 | 50.03 | 50.06 | 49.64 | 49.97 | **50.97** | 50.55 |
| instrs | 59.15 | 59.55 | 71.98 | 61.88 | 56.09 | 63.90 | 79.42 | 84.30 | 83.96 | 82.36 | 83.43 | **84.37** | 82.91 |
| office | 56.84 | 59.28 | 69.14 | 61.87 | 75.48 | 63.22 | 78.41 | 83.02 | 82.07 | 80.98 | 82.21 | **83.96** | 81.25 |
| outdoor | 55.81 | 57.39 | 56.54 | 58.23 | 56.17 | 57.12 | 70.11 | 74.43 | 75.03 | 72.52 | 75.03 | **75.22** | 73.60 |
| software | 49.70 | 50.45 | 49.97 | 50.73 | 53.30 | 49.36 | 51.90 | 52.57 | 52.25 | 52.51 | 52.25 | **53.24** | 52.77 |
| sports | 51.08 | 51.13 | 50.59 | 49.97 | 52.17 | 49.52 | 50.25 | 50.32 | 50.67 | 52.10 | 50.78 | **52.17** | 49.16 |
| tools | 55.95 | 56.04 | 78.35 | 62.50 | 22.32 | 41.96 | 81.25 | 86.61 | 86.92 | 85.11 | **87.50** | 87.50 | 86.30 |
| toys | 50.07 | 50.82 | 49.70 | 50.20 | 52.92 | 52.80 | 51.87 | 50.17 | 50.90 | 49.95 | 51.00 | **53.25** | 51.10 |
| video | 51.65 | 50.72 | 52.30 | 50.08 | 51.87 | 51.27 | 50.62 | 50.20 | 50.80 | 50.70 | 50.96 | **52.47** | 51.25 |

Table 8: Sentiment analysis accuracy [%] on unknown domains.

## A.7 SENTIMENT ANALYSIS

We show the full test and accuracy result for sentiment analysis in table 8.

