# OpenReview forum: "Learning Uncertainty for Unknown Domains with Zero-Target-Assumption"
_ICLR.cc/2023/Conference — ICLR 2023 poster_

### Official Review · Reviewer_xud3 · 2022-10-23

**Confidence:** 3
**Correctness:** 4
**Technical Novelty And Significance:** 2
**Empirical Novelty And Significance:** 3
**Recommendation:** 6

**Clarity, Quality, Novelty And Reproducibility:**

Clarity: The paper is very clear and well-written.

Quality: The quality of presentation, theoretical soundness of the technique, the experiments and of the significance of the results is good.

Originality: Within the context of the prior work presented in this paper for the task of NLP for optimal domain generalization, the proposed method seems novel. I am not an expert in this field of NLP and hence I am not deeply familiar with the specific literature in this field beyond what the authors have cited. However, the idea if using RL for optimal training data generation is not entirely new and has been used for example in prior works in the related field of computer vision in the following works:

(a) Ruiz et al, Learning To Simulate, ICLR 2019 (https://openreview.net/forum?id=HJgkx2Aqt7)
(b) Mishra et al, Task2Sim: Towards Effective Pre-training and Transfer from Synthetic Data, CVPR 2022 (https://openaccess.thecvf.com/content/CVPR2022/papers/Mishra_Task2Sim_Towards_Effective_Pre-Training_and_Transfer_From_Synthetic_Data_CVPR_2022_paper.pdf).

Reproducibility: Good! The authors have provided code (which I did not run) and have described the method is sufficient detail.



**Strength And Weaknesses:**

Strengths:
Novelty: The proposed method addresses an open under-researched and impactful problem of optimal generalization from seen to unseen domains. It presents the novel and intuitive insight of maximizing the entropy of the training dataset to improve generalization on target tasks. The paper also proposes a straight-forward and simple method based on RL to maximize this uncertainty measure.

Soundness: All aspects of the proposed method are theoretically sound and correctly motivated.

Clarity: The paper is very-well written and very clear.

Experiments: The proposed method shows significant and impressive performance improvements over all previous methods that it is compared against in the paper, and even those that use target datasets on four different datasets. The authors have also clearly empirically ablated their design choices for the type of RL algorithm and the uncertainty measure to use.

Weaknesses:
1. One obvious weakness of the proposed method is its high computational cost, which in turn stems from the use of the RL algorithm. RL is widely known to be sample inefficient. The use of the SAC versus the A2C algorithm mitigates this somewhat in the proposed design. While the authors allude to this this briefly in their paper, this weakness could be acknowledged and discussed more in the paper.

2. Another widely used strong baseline method for optimal sample selection with known target datasets is the following: Ren et al., Learning to reweight examples for robust deep learning, ICML 2018. However, the authors have not compared against it. It would have been interesting to see this comparison.

3. The proposed method could be made more impactful if the authors had shown it to also be applicable to other domains besides NLP, for example to vision as well.

**Summary Of The Paper:**

This paper proposes a method for optimal training set selection with the goal of maximizing generalization to multiple unknown target domains for NLP tasks. One of the goals of the method is to perform data selection on the training set only without knowledge of any target domain. To achieve this, the paper proposes a Maximum-Entropy Rewarded Reinforcement Learning (MERRL) framework, which seeks to maximize the entropy of the training data. The proposed method rests on the key assumption that the optimal training data for generalizing to many different tasks simultaneously is one that maximizes the training data's entropy and hence is not "latched on" to any one specific domain. The authors experiment with two RL algorithms (A2C) and SAC and two measures of entropy: observational entropy (OE) and prediction entropy (PE) and find SAC combined with PE to be be optimum. On four NLP benchmarks the authors show that their proposed approach outperforms several previous approaches for this task, including ones that specifically use the target dataset for optimal training set selection.

**Summary Of The Review:**

Overall, I am leaning positive about this paper based on the material presented in it with the caveat that I am not an expert in the field of NLP and hence not familiar with its most current literature. I am aware of closely related works from the field of computer vision that have previously employed ideas of using RL or meta-learning for optimal training set selection/synthesis. I would like to hear from the authors about how they position their method within the context of these existing closely related-works in the vision domain; what their thoughts are on the limitations of using RL for their task; and how generalizable they think their technique would be to other problem domains besides NLP.

---

> ### Author Response · Authors · 2022-11-16
> **Added related work**
>
> We thank the reviewer’s valuable comments and recognition of our contribution. We add the following clarifications to address the reviewer’s confusion:
>
> 1. “One obvious weakness of the proposed method is its high computational cost,...While the authors allude to this this briefly in their paper, this weakness could be acknowledged and discussed more in the paper. ”
>
> * We agree that the RL has high computational cost. One of our contributions is to introduce SAC into the data selection framework to reduce the sample complexity. Empirically, SAC could achieve a similar average reward level as baseline (VPG) within ten-times shorter time (~2000 steps VS 20000 steps). Moreover, one direction of future work is to incorporate the few-shot meta-RL to further reduce the computational cost.
>
>
> * We have added this acknowledgement of high computational cost and relevant discussions into the revised paper in section *Time* of 3.2 Analysis:
>  “In practice, training with all in-domain data in sentiment analysis takes 131 seconds while selecting data with SAC-OE takes 1394 seconds ($\mathcal{T}_{budget}=2000$ and $\mathcal{T}=140$) on one Tesla V100 GPU, which is roughly ten times faster than the baseline VPG to achieve similar average reward. That being said, the problem of computational cost for MERRL remains to be solved and we leave this as future work.”
>
> ____
>
> 2. “I would like to hear from the authors about how they position their method within the context of these existing closely related-works in the vision domain”
>
> * We thank the reviewer for bringing up many interesting related works in the field of computer vision. We examine the mentioned related works and add a paragraph of related works in computer vision and other fields:
> “Beyond the field of NLP, data selection Killamsetty et al. (2021); Durga et al. (2021), data augmentation Volpi et al. (2018); Zhang et al. (2020); Zhou et al. (2020) and data generation Ruiz et al.(2018); Qiao et al. (2020); Mishra et al. (2022) have been widely used in vision, medical and general regression tasks. These methods either utilize augmentation operations on images (e.g, flipping, scaling and cropping), or focus on one specific goal (i.e. image style transfer), or include generative models for adversarial training. Thus, it requires further consideration on how to generalize to text data when adapting these methods to NLP tasks. Our method puts an emphasis on both characteristics of text data and general prediction entropy which could be directly generalized to other fields. Another relevant and emergent line of work is data pruning, which aims at selecting a minimum subset of training data to reduce training costs Sorscher et al. (2022); Yang et al. (2022), or to enhance adversarial robustness of the model Kaufmann et al. (2022).”
>
>
>
>
> * In fact, the goal of the mentioned works is to generate new data, which increases the size of the training set, while we address the problem from a different angle by utilizing existing sample dependency among the current dataset. It is also worth noting that in NLP, using data generation requires consideration since it only works when strong features exist in the original dataset as shown by Jha 2020 (https://arxiv.org/abs/2004.15012).
> ____
>
> 3. “how generalizable they think their technique would be to other problem domains besides NLP.”
> * The generalization of entropy-based data selection is our on-going work and is an exciting direction of future work. In fact, our introduced prediction entropy could be directly used to optimize image datasets. However, there still exists the challenge of measuring observation entropy for image data, which is less addressed in previous literature.

---

> > ### Comment · Reviewer_xud3 · 2022-11-28
> > **Response to Authors**
> >
> > I thank the authors for answering my questions and for making appropriate adjustments to paper to address several of them. However, the lack of comparisons to the previous method proposed by Ren et al, on learning to re-weight training samples was a major concern of mine, which the authors have not addressed. This concern is also echoed in wzkA's comments about the lack of justification for the use an RL-based approach to optimize entropy versus using other more efficient approaches such as meta-learning. Hence, I have lowered by score.

---

> > > ### Author Response · Authors · 2022-11-30
> > > **Response to the lack of comparison to the previous method**
> > >
> > > Thank you for starting a discussion and highlighting your concern. We would like to argue that Ren et al. is not a relevant baseline. Ren et al. targeted class imbalance and label noise problems while we aimed at achieving generalization to unseen test domains. We have elaborated on Ren et al. later in the comment to have a more informed discussion. Regarding the lack of comparison, we would also like to highlight that we have incorporated six baselines in our work. Moreover, we agreed with the reviewers' comment on entropy as a baseline and we have added it in the revised manuscript (Table 8 in Appendix, column “Threshold”). Our method (average test accuracy 64.78 among 21 test domains) outperforms the “Threshold” (average 57.51).

---

> > > ### Author Response · Authors · 2022-11-30
> > > **Describing the work of Ren et al.**
> > >
> > > Here, we would like to summarize the work of Ren et al. Ren et al. addresses the problem of training set bias, specifically the class imbalance and label noise problems. Their motivation is that, for noisy label problems, examples with smaller training losses are preferred, however, for data with imbalance classes, examples with higher training losses are prioritized. To solve this contradiction, they proposed a meta-learning objective which is to find the weights of training samples that could minimize the validation loss. They experiment on class imbalance data and label corrupted data to show their method is effective at addressing training set bias. Kindly note that they did not run experiments on out-of-domain data since this is not the premise of their work. Moreover, their method makes use of an in-domain unbiased clean validation set during training (see Section 1, last paragraph in Ren et al).
> > >
> > > In contrast, we aimed at achieving generalization which is measured as the performance on a large set of unseen test domains. We select a subset of the training data with the most uncertainty while they calculate the weight of the training samples using an in-domain validation set.

---

> > > ### Author Response · Authors · 2022-11-30
> > > **Comparison with Ren et al.**
> > >
> > > Nevertheless, we implemented Ren et al. and evaluated its performance on the target domains. On average, the performance of Ren et al. is lower by 11.26 points than our method. It is even lower than the sentence-wise entropy-based selection (Table 8, “Threshold”). The low performance of Ren et al. is attributed to the fact that the method is not designed to improve generalization to unseen domains.
> > >
> > > In the following, we provide domain-wise comparison results of Ren et al. and our work:
> > > (our work vs. Ren et al.):
> > > Auto (78.53 vs. 63.58),
> > > Beauty (67.00 vs. 62.76),
> > > Food (81.98 vs. 68.12),
> > > Instruments (84.37 vs. 70.73),
> > > Office (83.96 vs. 68.63),
> > > Computer (68.46 vs. 64.14),
> > > Tools (68.46 vs. 64.14),
> > > Phones (62.99 vs. 57.08),
> > > Grocery (73.95 vs. 61.53),
> > > Jewelry (76.95 vs. 64.59),
> > > Outdoor (75.22 vs. 65.22)
> > >
> > > We are happy to discuss it further and any concerns you may have regarding our work. We appreciate your time.

---

### Official Review · Reviewer_wzkA · 2022-10-24

**Confidence:** 2
**Clarity, Quality, Novelty And Reproducibility:** The writing is good while the novelty…
**Correctness:** 4
**Technical Novelty And Significance:** 2
**Empirical Novelty And Significance:** 2
**Recommendation:** 5

**Strength And Weaknesses:**

Strengths:
In INTRODUCTION, the authors first proved the relationship between `training set entropy’ and `f1 score and oov’ through experiments, and explained the importance of training set entropy to model generalization. The motivation and logic of the paper is clear.

Weaknesses:
The authors only proved the role of entropy in selecting data, but this paper does not elaborate on the motivation and advantages of introducing complex reinforcement learning to train a policy network.

Further Comments:
1.	The authors use training set entropy as a reward to train a policy network for data selection. How is it different from directly using entropy and selecting data through threshold? What are their advantages and disadvantages?
2.	In the code provided in the supplementary materials, the policy network is first trained on the data set to be selected. Does the policy network need to be retrained on the data set to be selected each time? Considering the generalization of reinforcement learning, this will limit the universality of the algorithm.
3.	In Figure 5, the initial reward of SCA is much higher than A2C and VPG, and the trend is different from that of A2C and VPG. What is the reason for these phenomena?  Is the value of reward related to the quality of data selection?
4.	There are some clerical errors in the paper. For example, in second page, it should be “Pr(“nor”) = 1/2”. It is recommended that the author read the full text to correct such problems.


**Summary Of The Paper:**

In this paper, the authors introduce a novel framework, called Maximum-Entropy Rewarded Reinforcement Learning (MERRL), which can select training data to cover more possible queries that may appear in unknown worlds.

**Summary Of The Review:**

The authors only proved the role of entropy in selecting data, but this paper does not elaborate on the motivation and advantages of introducing complex reinforcement learning to train a policy network.

---

> ### Author Response · Authors · 2022-11-16
> **Clarifications on motivation of using RL and our novelty; Added experiments**
>
> We appreciate the reviewer for the valuable comments and we hope the following clarifications allow for a re-evaluation.
>
>
>
> 1. “The authors only proved the role of entropy in selecting data, but this paper does not elaborate on the motivation and advantages of introducing complex reinforcement learning to train a policy network.”
>
> * Unlike the conventional approaches that select *single* sentences with highest entropy, we select *a set of* sentences, i.e., the subset of all training sentences, with the highest entropy. The reason for doing so is to consider the **combination** of sentences, or **sentence (sample) dependencies**. Selecting a combination of sentences is a combinatorial optimization problem with exponential complexity (O(2^n)), thus, it is nearly impossible to try out all combination possibilities to derive an optimal solution. In contrast, reinforcement learning enables us to approximate an optimal solution to address this combinatorial optimization problem efficiently.
>
> __________
>
>
> 2. “The authors use training set entropy as a reward to train a policy network for data selection. How is it different from directly using entropy and selecting data through threshold? What are their advantages and disadvantages?”
> * Unlike the conventional approaches that select *single* sentences with highest entropy, we select *a set of* sentences, i.e., the subset of all training sentences, with the highest entropy. The reason for doing so is to consider the **combination** of sentences, or **sentence (sample) dependencies**.
> * We run an experiment for sentiment analysis by ranking sentences by sentence-level entropy and selecting sentences through a threshold and add the result to Table 8 in Appendix (column “Threshold”, the threshold is selected as the size of our SAC selected data). Our method (average test accuracy 64.78 among 21 test domains) outperforms the “Threshold” (average 57.51).
> * More specifically, one previous work Song 2012 (Entropy-based Training Data Selection for Domain Adaptation) shows selecting data by ranking sentences through sentence-level entropy cannot outperform random selection with statistical significance. In contrast, our work is novel at proposing observation entropy, which is the notion of set-level entropy considering sentence dependency, and prediction entropy, which trains a latent representation suitable for generalization.
>
>
>
> __________
>
> 3. “In the code provided in the supplementary materials, the policy network is first trained on the data set to be selected. Does the policy network need to be retrained on the data set to be selected each time?”
> * Our MERRL framework is designed to be applied in a realistic scenario where one training dataset is given and our task is to determine a best subset of data  for generalization. The generalization is about lowering error rates on the test set in different domains (it does not care about the training data although we select data as a way to achieve generalization). The algorithm is a paradigm that can be applied on any dataset. Each dataset/task should run this algorithm to its own selected data. Our experiments verified that on a number of OOD test sets our methods outperform the baselines.
>
> __________
>
> 4. “In Figure 5, the initial reward of SCA is much higher than A2C and VPG, and the trend is different from that of A2C and VPG. What is the reason for these phenomena?"
> * Different from A2C and VPG, SAC is an off-policy method, which means it updates the policy by randomly sampling a (state,action,reward) tuple from the replay buffer (step 7 in Figure1(a)). In contrast, the tuples A2C and VPG used to update the policy are the same, following a uniform transition function (see last paragraph in section 2.1). In a nutshell, the different tuples SAC used to update the policy are the reason leading to a different reward curve.
> __________
>
>
> 5. "Is the value of reward related to the quality of data selection?”
> * The absolute value of reward does not affect the quality of data selection, since it is related to the size of the batch (see 3.2 Batch size). However, the relative higher rank of reward under the same hyper-parameter setting indicates a better data selection quality. For example, in Figure2-left, at around 30000 timesteps, SAC and A2C achieve a higher average reward (1900) than VPG (1700), which reflects that the data selected by SAC/A2C is better than those selected by VPG.
>
> __________
>
>
> 6. “...clerical errors in the paper. For example, in second page, it should be “Pr(“nor”) = 1/2”
> * We thank you for pointing this typo out. We have revised it.

---

> ### Author Response · Authors · 2022-12-01
> **Thank you**
>
> Dear reviewer,
>
> Thank you again for your valuable feedback and for suggesting an important baseline. We have incorporated your feedback and have added the baseline. We look forward to hear from you.

---

### Official Review · Reviewer_aHZE · 2022-10-24

**Confidence:** 3
**Correctness:** 3
**Technical Novelty And Significance:** 3
**Empirical Novelty And Significance:** 3
**Recommendation:** 6

**Clarity, Quality, Novelty And Reproducibility:**


This paper is easy to follow. The authors provide the detailed information for reproduce the experiments.

**Strength And Weaknesses:**


Strength: In general this paper is clearly written and easy to follow. The experimental results seem to confirm the validity of proposed method.

Weakness: I suggest the authors include more justifications when selecting certain models of NLP tasks. For example, in sentiment analysis experiments, why the authors choose to use a CNN classifier as the model?

**Summary Of The Paper:**


In this paper, the authors propose to use a Maximum-Entropy Rewarded Reinforcement Learning framework to select training data for NLP tasks, the goal of which is to maximize generalization. The authors experiment with A2C and SAC and experimental results show that the proposed framework could outperform several baseline approaches.



**Summary Of The Review:**


I think the authors propose a novel method of using entropy in selecting data.

---

> ### Author Response · Authors · 2022-11-16
> **Added justifications for selecting certain NLP models**
>
> We thank the reviewer for the valuable comments. We hope the following clarifications address the reviewer’s concerns.
>
> “I suggest the authors include more justifications when selecting certain models of NLP tasks.”
>
> Our proposed framework, MERRL, is a model-agnostic data selection framework that can be applied to any NLP model. Our choice of using a particular model is motivated to either have a fair comparison with the previous work or to use a state-of-the-art model. For example, we use the CNN classifier for sentiment analysis to have a fair comparison with our baseline Liu et al. 2019 (the first line on page 7). For named entity recognition, we select the state-of-the-art model, BERT. For language modeling, we use the transformer architecture, which is the backbone of many state-of-the-art language models. The diversity of models used in the paper further supports the model-agnostic feature of our framework.

---

### Decision · Program_Chairs · 2023-01-20

**Decision:**

Accept: poster

**Justification For Why Not Higher Score:**

* some architecture blocks simply do not speak to a modern NLP audience (e.g., CNNs for sentiment classification, LM trained from scratch on selected data);
* the paper could nuance most uses of the word/acronym "NLP" in its narrative.

**Justification For Why Not Lower Score:**

* plausible and original premise, incorporated reasonably well into an algorithm for data selection
* elegant use of RL to circumvent combinatorial search
* overall, nicely executed paper
* the rebuttal addressed most criticism gathered in the first round

**Metareview: Summary, Strengths And Weaknesses:**


Strengths

* the paper has a clear motivation and addresses an important topic (data selection for out-of-domain generalisation);
* the paper is very clear and proposes a technically sound solution;
* their original method outperforms reasonable baselines across a few benchmarks (text classification, named-entity recognition, and language modelling -- in the sense of next word prediction with evaluation via perplexity).

Weakness

* RL is sample inefficient (I'd take that as a weakness only if some other sample efficient baseline existed, see remarks below) and details about the computation cost of the method were missing;
* Some unusual architectural choices (e.g., sentiment classification should be done on top of a large pretrained model; language models are not trained from scratch these days, rather, they would be fine-tuned on your selected data);
* NLP-only benchmarks;
* Benchmarks are not representative of NLP as a whole, yet the paper keeps referring to NLP as a whole. I don't think a single paper can touch upon enough NLP tasks to make a general claim about NLP, thus my advice is to nuance the language in the paper.

Remarks

* there were some reservations about RL (in particular it's sample inefficiency) that I am choosing to ignore in light of the rebuttal which further characterised the computational cost of the method;
* the reviewers requested two missing comparisons (against ranking on entropy and thresholding, and against Ren et al), both sufficiently addressed in the rebuttal;

The paper is executed reasonably well and I see no obvious technical flaws. Some reasons for rejection might be the unusual architecture choices (the paper's findings would be more readily available to a wider audience if experiments had exploited modern pretrained components throughout) and the overly general language around "NLP" when the investigation only covers 3 little corners of NLP.

**Note From Pc:**

if the above contains the word "oral" or "spotlight" please see: "oral" presentation means -> notable-top-5% and "spotlight" means -> notable-top-25%. As stated in our emails, we are disassociating presentation type from AC recommendations

**Summary Of Ac-Reviewer Meeting:**

No such meeting happened.